# Enabling accurate and early detection of recently emerged SARS-CoV-2 variants of concern in wastewater

Nicolae Sapoval ®[1], Yunxi Liu[1], Esther G. Lou[2], Loren Hopkins ®[3,4], Katherine B. Ensor[4], Rebecca Schneider[3], Lauren B. Stadler[2]✉ & Todd J. Treangen ®[1]✉

As clinical testing declines, wastewater monitoring can provide crucial surveillance on the emergence of SARS-CoV-2 variant of concerns (VoCs) in communities. In this paper we present QuaID, a novel bioinformatics tool for VoC detection based on quasi-unique mutations. The benefits of QuaID are three-fold: (i) provides up to 3-week earlier VoC detection, (ii) accurate VoC detection (>95% precision on simulated benchmarks), and (iii) leverages all mutational signatures (including insertions & deletions).

Wastewater monitoring is an invaluable tool for SARS-CoV-2 surveillance[1–4]. Despite multiple recent successes in VoC monitoring and detection from wastewater sequencing data[5–7], there are multiple challenges associated with the nature of the environmental data. Since wastewater represents a pooled sample of multiple hosts, it harbors a diversity of SARS-CoV-2 variants that are currently circulating in the population[1,2], including potentially previously unreported genotypes[8]. Variant detection and phasing are further complicated by uneven genome coverage[2] and environmental RNA degradation[9], which render phased assembly difficult[10]. Despite these challenges, detection of VoCs in wastewater samples is important for monitoring the emergence and spread of variants and informing public health response[7,9]. Current approaches for VoC detection in wastewater samples typically require sufficient depth and breadth of coverage of the variant genomes[5,11], and therefore depend on a large fraction of the sample representing the variant genotype[6], hampering early detection. Furthermore, most current approaches discard insertion and deletion (indel) information and only rely on single nucleotide variants (SNVs) associated with the VoC[5,11]. However, recently several new approaches have emerged that take indels into account[12,13], most notably COJAC[12] utilizes co-occurrence information for the mutation providing high specificity of detections. Finally, all approaches that rely on a database of previously collected SARS-CoV-2 genomes are biased by the contents of the database[14], which can lead to both false negative and false positive calls at the inference stage[15]. This issue can be further amplified when the underlying database is not scrutinized for potential metadata errors.

In this work, we introduce QuaID: a computational pipeline for analyzing SARS-CoV-2 wastewater sequencing data and inferring presence of VoCs, that leverages both SNV and indel data. We demonstrate the performance of our tool on real Houston wastewater data detecting Alpha, Delta, and Omicron VoCs. We also compare the performance of QuaID and state-of-the-art tool Freyja[5] on both real and simulated data. We demonstrate enhanced precision of QuaID in the simulated benchmark, and earlier detection of the Omicron VoC in Houston based on the real data. Thus, we demonstrate the benefits of using QuaID for the early and accurate detection of SARS-CoV-2 VoCs.

## Results

Between February 23, 2021, and May 5th, 2022 we collected, processed, and analyzed 2,637 wastewater samples from the fifth-most populous metropolitan area in the US: Houston, Texas. Samples were collected weekly from 39 wastewater treatment plants (WWTPs, Supplementary Table 1, Supplementary Figure 1) distributed throughout the city of Houston and servicing more than two million Houston residents[16]. During the study period, the VoC detection signal clearly reflected the three major variants that affected Houston - Alpha, Delta, and Omicron (Fig. 1b). QuaID was able to detect the Delta VoC two weeks prior to the first sequenced clinical sample in Texas (marked by star in Fig. 1c) and continued to provide detection signal for the four subsequent weeks after the first sequenced clinical sample (2021-04-05 to 2021-05-03). In contrast, Freyja reliably picked up the Delta signal only once the VoC became more prevalent. Similarly for the Omicron VoC, QuaID detected

[1]Department of Computer Science, Rice University, 6100 Main Street, Houston, TX 77005, USA. [2]Department of Civil and Environmental Engineering, Rice University, 6100 Main Street, Houston, TX 77005, USA. [3]Houston Health Department, 8000 N. Stadium Dr., Houston, TX 77054, USA. [4]Department of Statistics, Rice University, 6100 Main Street, Houston, TX 77005, USA. ✉e-mail: lauren.stadler@rice.edu; treangen@rice.edu

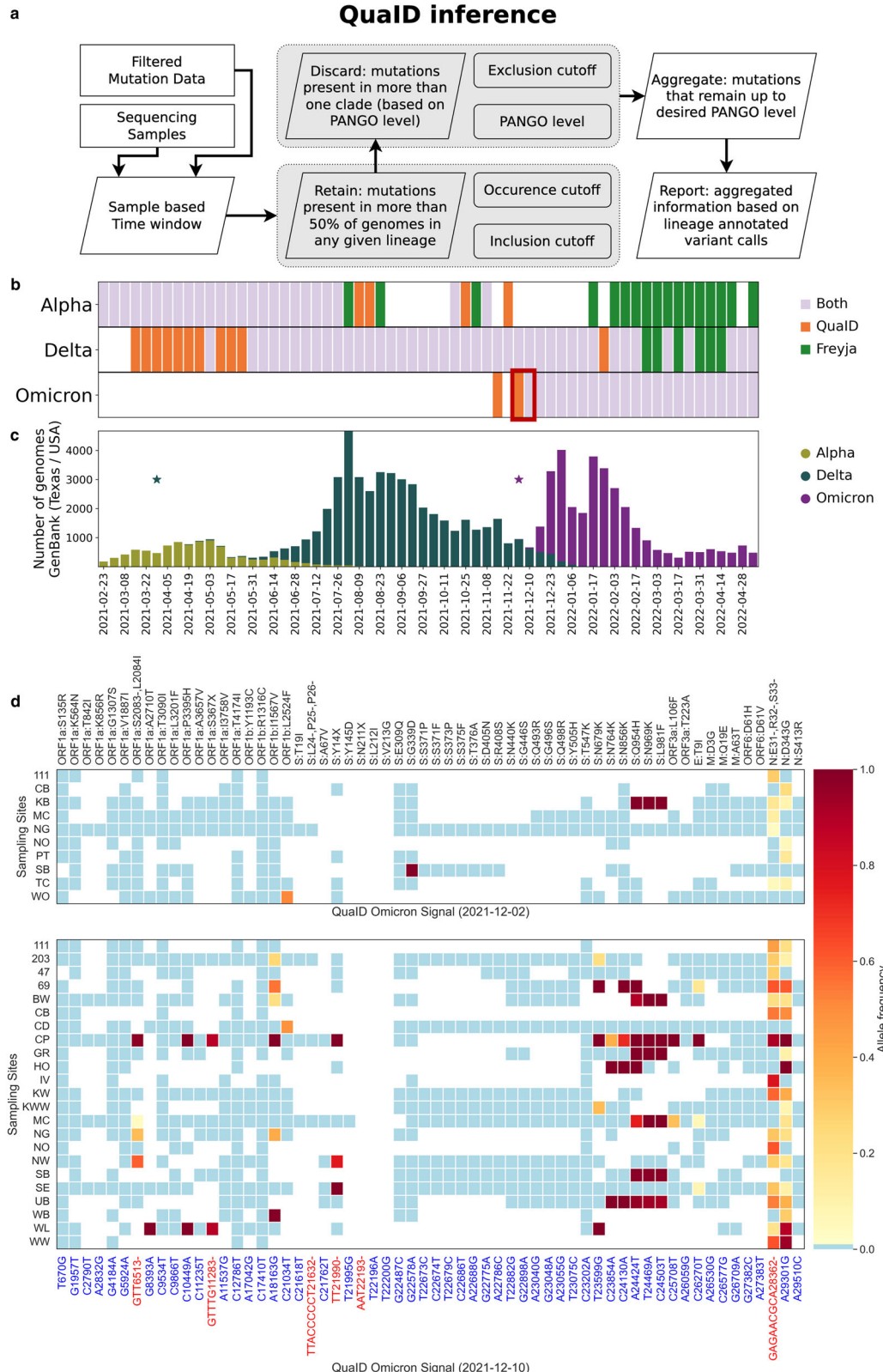

**Fig. 1 | Detection of Alpha, Delta, and Omicron VoCs in Houston, TX waste-water. a** QuaID VoC inference process overview. Parameters that affect described subroutines are provided in the rounded rectangles. **b** Early detection of the emerging variants of concern in Houston wastewater provided by QuaID and Freya pipelines. For Omicron and Delta variants, QuaID provided earlier detection. Each week is presented as the aggregate signal from the 39 WWTPs with detections being reported if at least 2 WWTPs had any QuaID signal or had any non-zero abundance of

the VoCs reported by Freyja. **c** Variant prevalence in the clinical data over the study period obtained from GenBank and restricted to Texas. Stars indicate the first occurrences of a Delta variant genome (dark green) and an Omicron variant genome (purple). **d** Heatmaps of WWTPs with detected Omicron variant quasi-unique mutations the week of December 2nd, 2021 (top) and December 10th, 2021 (bottom) in Houston. Blanks indicate lack of sequencing data, blue color indicates no mutation detected, and the gradient shows the allele frequency for detected mutations.

the presence of the variant in wastewater two weeks prior to the first clinical sample collection date, while Freyja required an additional week after the first clinical sample to detect Omicron presence. Furthermore, QuaID reported detections of BA.2 sublineage of Omicron starting 2021-12-16, and BA.5 sublineage of Omicron starting 2022-01-13 (Supplementary Data 1). Consistent detections of these two major sublineages of Omicron have been reported in our data throughout 2022.

We further investigated the early detection of the Omicron variant in Houston wastewater by visualizing a heatmap of variant calls (Fig. 1d) and examining the multiple sequence alignment (MSA) of SARS-CoV-2 Omicron variant genomes available on GISAID[17] in early December 2021. We observed 50% (5 out of 10) of the samples with Omicron presence for the week of December 2nd, 2021 contained the 9 bp deletion (N:DEL31/33), which is a stable mutation (95.1% prevalence among all Omicron genomes[18]) for the Omicron variant (Fig. 1d). Since the current version of Freyja relies on the UShER[19] phylogenetic tree for its designation of mutational signatures, no deletions are used in the inference process, highlighting one of the reasons for the delayed detection of the Omicron variant. In the subsequent week, December 10th, 2021, when both Freyja and QuaID reported the presence of the Omicron variant in the wastewater, the N:DEL31/33 mutation was present in 16 of 23 sites with detections (Fig. 1d), and for one of the samples with no deletion there was no coverage in the region flanking the deletion (Fig. 1d, Sampling Site SB: Sims Bayou North).

To further examine the sensitivity of the QuaID and Freyja to degradation of the sequencing data, we conducted simulated experiments following three protocols motivated by the following empirical observation from the wastewater data. In the real wastewater sequencing data, 37.7% of all samples had less than 25% of the SNVs associated with the Omicron VoC via UShER barcodes covered by at least one read (Supplementary Figure 3B), and 24.4% of samples had less than 10% of all Omicron-associated SNVs with at least one read. The first simulation protocol (a) retained a percentage of SNVs at random from a simulated sample. Thus, we constructed three simulation scenarios with each retaining 10%, 25%, or 50% of all SNVs. Our results show that due to the inclusion of deletion information in the inference process, QuaID remained sensitive even when only 10% of all SNV calls were retained, while Freyja required at least 50% of the calls to be included to reliably detect the VoC presence. In particular, when only 10% of all SNV calls were retained, QuaID still detected the presence of Delta and Omicron VoCs reliably, and Alpha and Gamma VoCs sparsely, while Freyja failed to estimate the abundance of any of the VoCs (Alpha, Delta, Gamma, and Omicron) present in the simulated samples (Fig. 2a, Supplementary Figures 4A–7A). Furthermore, when 25% of all SNVs were retained, QuaID identified the present VoCs in the majority of the simulated samples, while Freyja provided sparse detection in the samples dominated by a single VoC (Fig. 2b). Finally, when 50% of all SNVs are retained, Freyja detected most of the VoCs present in the samples, and in several instances recovered the correct relative abundance. However, even in this scenario 8 Omicron-dominated samples failed to be correctly identified by Freyja, while QuaID correctly inferred the presence of the VoC. Additionally, we observe that the stability of the coverage for the N:DEL31/33 is further empirically supported by our data, which indicated that among all samples more than 61% had at least 10 reads that covered the bases immediately flanking the deletion (Supplementary Figure 3C). Next, we considered a simulation protocol (b), where SNVs were resampled as Bernoulli trials based on the coverage of a real guide sample. The results from these experiments support the robustness of QuaID detections (Supplementary Figures 8–31).

Finally, we devised a simulation protocol (c) which resampled simulated sequencing reads based on a coverage profile guide from a real sample. In total, we generated 32,448 simulated samples for this protocol and compared the performance of QuaID, and Freyja. The results of this comparison aggregated over all 32,448 samples are presented in Table 1. In this set of simulated experiments, QuaID had the highest precision for all VoCs in consideration, further supporting our confidence in early detection of Delta and Omicron VoCs in Houston wastewater data. Freyja achieved a good balance between precision and recall as indicated by high F1 score. We hypothesized that the high precision of QuaID was likely due to the combined variant caller output used in the pipeline. To test this, we conducted an experiment in which QuaID used only iVar output for its analyses. As expected, we found that QuaID's recall matched or in some cases surpassed that of Freyja at the cost of lower precision when using iVAR only (Supplementary Figure 32A, B).

## Discussion

Wastewater monitoring of the emergence and spread of SARS-CoV-2 variants offers unique benefits based on the early detection of the variant arrival prior to the clinical data[3, 4,20,21], and broad surveillance coverage of the population[4,20]. QuaID offers a solution for accurate and early VoC detection with tolerance for degraded data for VoC detection using wastewater SARS-CoV-2 sequencing data. In comparison to one of the leading tools for analyzing SARS-CoV-2 wastewater sequencing data for VoC detection, QuaID demonstrated superior sensitivity to the empirical data, and higher precision in simulations. This is particularly important given that the underlying sample quality and the depth and breadth of coverage of amplicon sequencing data can vary widely across samples[16]. Furthermore, the ability to leverage indel information in the inference process makes QuaID overall more robust than approaches that rely solely on SNVs. However, assessing the impact that analysis of indels provides in the task of VoC detection requires more extensive evolutionary modeling.

QuaID also has some limitations in its design. Since QuaID is an early detection tool it does not perform full phylogenetic placement of reads, which in cases when data quality is high can provide a more robust representation of the sample's lineage composition. Additionally, since our main goal was accurate and early detection of emerging variants in scenarios where the underlying mutational signal is low, QuaID treats each observed mutation as an independent event, and hence is not in its current form suited to perform relative abundance estimates. Analogous to the other tools for variant detection, QuaID depends on the database of known lineages, and hence cannot detect an emerging lineage that has not yet been designated as novel. However, the ability to track quasi-unique mutations for all known lineages combined with the variant calling in longitudinal samples can be extended in future work to also enable detection of recurrent novel mutations, and hence putative novel lineages. This is also important if the rate of clinical sequencing declines, leading to less densely sampled databases and hence potential for missing lineages in the available clinical sequencing data.

We envision QuaID to be one of several tools routinely employed in wastewater monitoring efforts. For example, QuaID could be used in parallel with Freyja and COJAC to achieve high sensitivity for detecting emerging variants using QuaID, relative abundance estimates for the dominant circulating variants using Freyja, and high specificity confirmations for detection events using COJAC. Furthermore, future work on extending the framework of QuaID and other tools to other pathogens that can be detected in the wastewater can enable sensitive and continuous environmental monitoring beyond the COVID-19 pandemic. Finally, given the multitude of technical challenges posed by the inherent variability and quality of wastewater sequencing data, we believe that establishing extensive sets of simulated and synthetic datasets that emulate challenges in variant calling in wastewater samples is required to further expand our understanding of how RNA degradation, sample preparation and storage techniques, and sequencing protocols affect the downstream data and analyses.

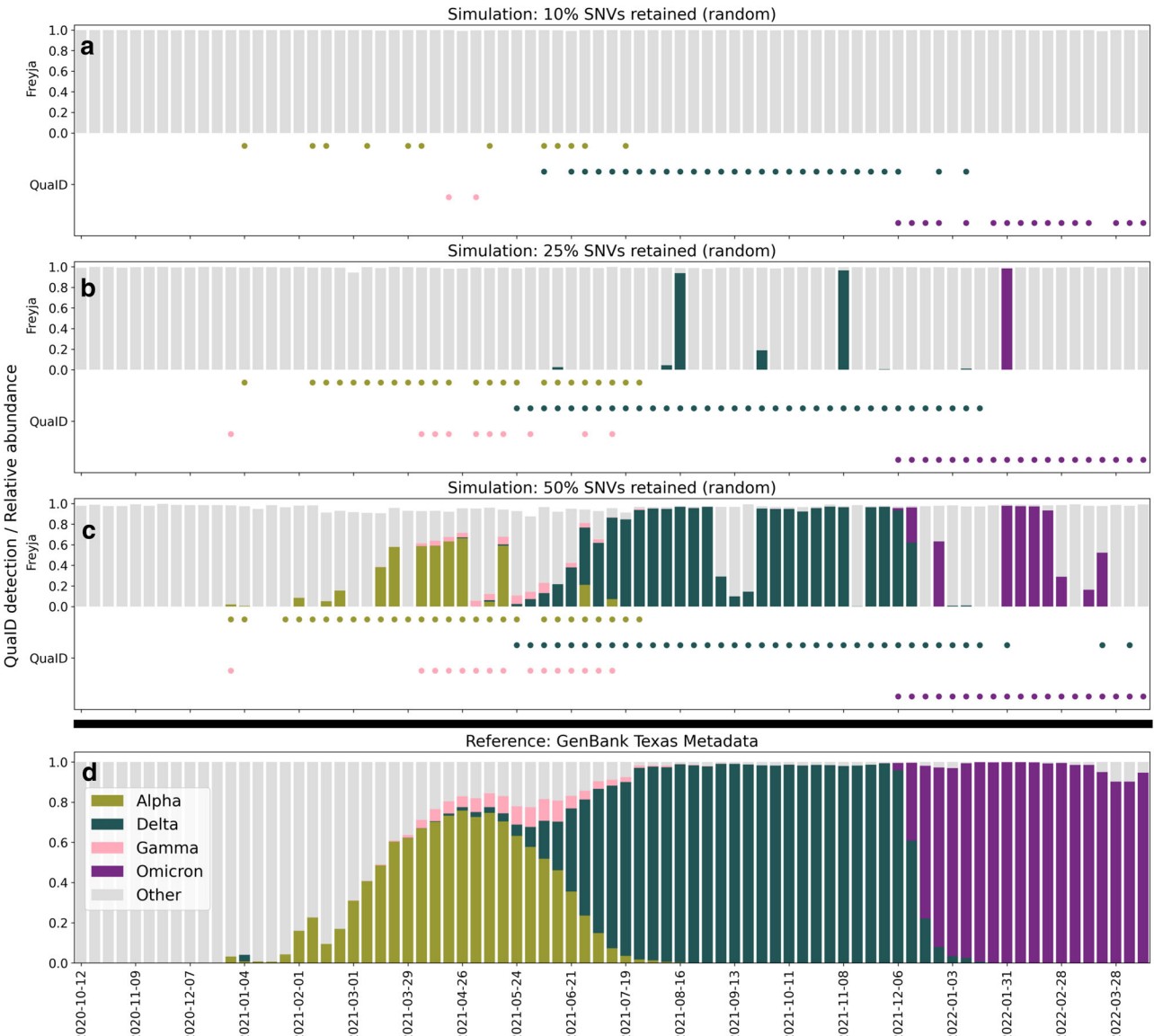

**Fig. 2 | Detection of VoCs in simulated data at various levels of SNV dropout.** **a** Freyja relative abundance estimates and QuaID detection signal on simulated data from GenBank (USA/TX) with 10% of all SNVs retained at random. Freyja is unable to detect any of the four (Alpha, Delta, Gamma, Omicron) VoCs. **b** Freyja relative abundance estimates and QuaID detection signal on simulated data from GenBank (USA/TX) with 25% of all SNVs retained at random. Freyja sparsely detects major VoCs (Delta, Omicron). QuaID detections are less sparse for all VoCs. **c** Freyja relative abundance estimates and QuaID detection signal on simulated data from GenBank (USA/TX) with 50% of all SNVs retained at random. **d** Metadata from GenBank (USA/TX) showing the fraction of genomes belonging to different VoCs for any given week. In this simulated experiment, the fractions shown correspond to true relative abundances in the simulated mixture.

## Methods

### Wastewater sample collection, RNA extraction, and sequencing

Houston Water collected and provided weekly 24-hour time-weighted composite influent (raw wastewater) samples from 39 wastewater treatment plants (WWTPs) in Houston covering a service area of approximately 580 miles[2] and serving over 2.3 million people. In total, 2637 samples were analyzed. Untreated wastewater samples were collected from the influent channel using refrigerated 24-hour composite samplers at each wastewater treatment plant. The autosamplers collected an aliquot of wastewater (200 mL) every hour over 24 hours. After sample collection, samples were placed on ice and transported to Houston Water's laboratory, aliquoted into 250 mL and 500 mL bottles, and transported on ice to Rice University (Rice) for processing. Houston Water provided the influent flowrates for each WWTP corresponding to the 24-hour sampling period. SARS-CoV-2 was concentrated in wastewater samples using an electronegative filtration

method as previously described[22]. We followed the same RNA extraction as described in prior work[16]. Namely, RNA extraction was performed using a Chemagic™ Prime Viral DNA/RNA 300 Kit H96 (Chemagic, CMG-1433, PerkinElmer) with the PerkinElmer viral RNA/ DNA purification protocol and reagents. Sequencing was performed by the Houston Health Department laboratory and several different sequencing kits, library preparation kits, and primer panels were used to amplify SARS-CoV-2 genomes in wastewater samples (Supplementary Table 2). Changes in sequencing protocols were made based on availability of reagents and to update primers that were better optimized for emerging VoCs. cDNA was generated via reverse transcription using the Superscript IV first-strand synthesis system (ThermoFisher Scientific, 18091050) following the manufacturer's protocol. Each sample library was then quantitated, normalized, pooled, and diluted. Finally, sequencing was performed on an Illumina MiSeq instrument using the kit and cycling conditions specified in

**Table 1 | Precision (number of true positives divided by the sum of true and false positives), recall (number of true positives divided by the sum of true positives and false negatives), and F1 score (harmonic mean of precision and recall) for QuaID and Freyja on the data simulated under protocol (c) (total number of samples analyzed, *n* = 32,448)**

| Variant | Tool | Precision | Recall | F1 score |
|---------|------|-----------|--------|----------|
| Alpha | QuaID | 0.954 | 0.517 | **0.671** |
|  | Freyja | 0.847 | 0.555 | 0.670 |
| Delta | QuaID | 0.979 | 0.532 | 0.689 |
|  | Freyja | 0.747 | 0.663 | **0.702** |
| Gamma | QuaID | 0.999 | 0.343 | 0.511 |
|  | Freyja | 0.686 | 0.414 | **0.516** |
| Omicron | QuaID | 1.0 | 0.472 | 0.642 |
|  | Freyja | 0.859 | 0.614 | **0.716** |

Bold font indicates highest F1 score.

Supplementary Table 2. Additional details on WWTP sample sites, methods regarding sample collection procedures, and quantification of SARS-CoV-2 in wastewater samples can also be found in our previous publication[16]. Estimates of the viral load are provided by the Houston Health Department following the same methodology as outlined in the SARS-CoV-2 Wastewater Monitoring Dashboard[23].

### Amplicon sequencing data processing
We processed the MiSeq paired-end data through a standard sequence of steps consisting of quality control report generation (FastQC[24], default parameters), quality and adapter trimming (BBDuk[25], quality trimming both ends of the read with threshold 15, and trimming standard PhiX adapter sequences), read mapping (BWA MEM[26], default parameters), and primer site soft clipping (iVar[27], ARTIC v3[28] primer scheme, minimum quality threshold 15). The summary overview of the whole processing pipeline is presented in Supplementary Figure 2A.

### Variant calling
We obtained two sets of variant calls for each sample: one with iVar[27] (minimum quality 20, minimum allele frequency 0) and the other with LoFreq[29] (after adjusting quality scores for indel calling with the 'lofreq indelqual–dindel' call, variant calling parameters are set to default). Both variant callers were configured to output all variant calls regardless of the allele frequency. We then used custom Python code to perform a variant call merge-and-filter operation which retained only those variant calls that were supported by both variant callers and had an allele frequency equal or above the user-defined threshold (default: 0.02) according to at least one of the two variant callers (while allele frequency estimates are typically close between the two variant callers differences of <0.01 occur).

### Sequence database sanitation
Prior to the subsequent analysis, we used metadata obtained from GISAID website to filter out sequences that were marked as incomplete or that had an associated host other than *Homo Sapiens*. Additionally, VoCs with a large amount of clinical sequencing data available (Alpha, Delta, Omicron) is more prone to human error in the metadata entries. Thus, we implemented a filter that removed any genomes: annotated as Alpha with submission date prior to September 3rd, 2020, annotated as Delta with submission date prior to March 1st, 2021, and annotated as Omicron with submission date prior to September 1st, 2021 (first detection dates based on cov-lineages.org VoC reports). Finally, we excluded all recombinant PANGO lineages[30] (X*) from the analysis.

### Mutation database construction
We used the pre-generated MSA file from GISAID to extract all mutations using vdb[31] in nucleotide mode with ambiguous bases included. We then trimmed the resulting list of mutations using the vdb trim command. Finally, we linked the resulting mutation list with the metadata based on the genome accession IDs, and the resulting data were aggregated by week and lineage through custom Python code. Additionally, any SNVs that resulted in an ambiguous base call (e.g. N, W, S, etc.) were removed from the database (summary view provided in Supplementary Figure 2B). The resulting data were used as the mutation tables to calculate prevalence of mutations in PANGO lineages[30] over a user-defined time window (default: 4 weeks).

### Quasi-unique mutations
For each lineage and mutation combination, the prevalence of the mutation occurring in the corresponding lineage's genomes was calculated and then converted to a fraction of all genomes assigned to the lineage. Mutations that appeared in >50% of all genomes for a single lineage (i.e., not appearing in any other lineage at 50% or more) were considered quasi-unique for that lineage. The above choice of inclusion (what fraction of genomes in the lineage must have the mutations) and exclusion (what fraction of genomes in any other lineage precludes the mutation from being selected) corresponds to the definition of a consensus genome but can be modified to arbitrary values by the end user. Setting stricter thresholds (requiring more of the target lineage genomes to have a mutation) will lead to smaller sets of quasi-unique mutations of high confidence, trading of sensitivity for specificity. Furthermore, since often we want to report detections at a higher level (e.g., any Omicron sub-lineage as opposed to a specific leaf node like BA.2.1) when determining which genomes are used for the exclusion rule, all the genomes that come from the same sub-clade at a fixed level (default: 4) in PANGO hierarchy are omitted from the exclusion check. Thus, mutations common to BA.1 and BA.2 can still be considered as quasi-unique for the Omicron VoC. Note that since vdb reports out deletions and we only filter out ambiguous SNVs, a quasi-unique mutation can be a deletion. Additionally, in order to reduce potential noise from rare lineages, we omit any lineages which have less than a user-defined count of genomes (default: 2) within the designated time window. An overview of these processes is presented in Supplementary Figure 2C. Finally, for each quasi-unique mutation QuaID estimated its predictive power as the posterior probability of observing a particular lineage given the observed mutation. Formally, for a lineage of interest $l$ and the quasi-unique mutation $m$ we computed $P(l|m)$ using Bayes' theorem $P(l|m) = \frac{P(m|l)P(l)}{P(m)}$. We let $P(m)$ to be the ratio of the number of genomes with the mutation $m$ observed to the total number of genomes in the database. Next, we let $P(l)$ to be the ratio of the number genomes belonging to the lineage $l$ and the total number of genomes. Finally, we let $P(m|l)$ to be the fraction of genomes in the lineage $l$ containing the mutation $m$. While we did not provide any filtering based on the estimated predictive power of the quasi-unique mutations, these probabilities can be used in the downstream analyses to improve the interpretations of the detection signal provided by QuaID.

### Mutational signature aggregation
Since the PANGO lineage hierarchy continuously expands potentially introducing new sub-levels for any lineage, it is useful to aggregate quasi-unique mutations into sets that correspond to a node at a fixed level of the hierarchy. For example, Omicron variant is defined as any descendant of B.1.1.529 PANGO lineage, and thus Omicron corresponds to level 4 in the hierarchy. When aggregating quasi-unique mutational signatures up to a given level, we took the union of all descendant lineage quasi-unique mutation sets. Note that the aggregation step always uses the same level of the hierarchy as the exclusion step of the quasi-unique mutation set construction.

## Variant of concern detection

Given a wastewater-based sequencing sample collected on a given date $D$, we constructed the corresponding sets of quasi-unique mutations in the time-window prior to and including weeks up to date $D$ (in case when there is no database information for the week(s) immediately preceding the target date $D$, the last available time-window was used). Then we merged the filtered set of variant calls for the sample with the quasi-unique set of mutations with the key set to the nucleotide change. We also filtered out any SNVs from the sample that result in synonymous mutations. Once the combined data is obtained, we reported for each sample the total combined allele frequency and total count of observed quasi-unique mutations, as well as the total possible number of quasi-unique mutations for the variants of interest at the desired level. Additionally, we reported what percentage of the quasi-unique mutation sites had coverage (with deletions being evaluated based on the genomic positions flanking the deletion) in order to distinguish between the no detection and no coverage scenarios. A detailed description of the outputs provided by QuaID can be found in Supplementary Table 3, and in the GitLab repository README file.

## Benchmarking tool performance with simulated data

We have considered three simulation protocols in order of increasing complexity: (a) random SNV dropout model, (b) coverage template-based SNV resampling, and (c) coverage template-based read resampling. In all three cases the base dataset was constructed from SARS-CoV-2 sequences deposited into NCBI GenBank before 15 April 2022, and collected between 11 April 2020, and 15 April 2022. Sequences were downloaded and grouped based on the collection week, yielding a total of 104 groups. For each group read data was simulated using ART[32] short-read simulator. Each genome in the set representing a week was sampled with the same coverage, thus lineages with more representatives in each time frame yielded a higher amount of reads for that sample. Then the reads were processed identically to the regular pipeline processing. For the simulation protocol (a) after the variant calling was performed and results of LoFreq and iVar calls combined, we randomly retained 10%, 25%, or 50% of all SNVs that were called. In protocol (b) we augmented this process by introducing coverage templates. We selected 4 weeks of real data (corresponding to 06/14/2021, 07/12/2021, 11/15/2021, and 12/16/2021) and for each week we considered all 39 WWTPs and their corresponding sample coverage profiles. Given a coverage profile we performed a SNV resampling procedure directly on combined variant call files. The procedure consisted of changing the total depth of coverage at a SNV position to that in the coverage profile and if the resulting total depth was below 10 the SNV was removed, otherwise, the AF of the SNV was used as the probability of success in Bernoulli trial, and the fraction of successes from the total depth number of trials was set as the new AF for the SNV. However, since both protocols (a) and (b) operated at the level of variant calls we could not use them for a complete comparison. Thus, for simulation protocol (c) we directly resampled reads in order to approximate coverage in the template profile. Template profiles were identical to the ones used in protocol (b). Given a coverage profile, we evaluated the coverage in positions 80 base pairs away from the 3' and 5' ends of the ARTIC v3 amplicons and used these values as target coverages. In order to build a sample, we sampled without replacement reads from the simulated data that overlapped at least one of target positions until either the target coverage, as defined above, was achieved or there were no more reads left to sample. The pipeline was then re-run with the modified read mapping files and the three tools benchmarked.

## Reporting summary

Further information on research design is available in the Nature Portfolio Reporting Summary linked to this article.

## Data availability

Sequencing data used in this manuscript is available at SRA via BioProject accession PRJNA796340. SARS-CoV-2 genomes used for the simulation dataset construction are available via GenBank (https://www.ncbi.nlm.nih.gov/labs/virus/vssi/#/virus?SeqType_s=Nucleotide&VirusLineage_ss=taxid:2697049). SARS-CoV-2 multiple sequence alignments and their associated metadata used in the database construction are available via GISAID (https://gisaid.org). Minimal set of data to run a demo of QuaID is available via Box (https://rice.box.com/v/QuaID-example-data).

## Code availability

Code developed and used in this study[33] is available on GitLab: https://gitlab.com/treangenlab/quaid, and has also been deposited on Zenodo: https://doi.org/10.5281/zenodo.7803146.

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

## Acknowledgements
The authors thank the GISAID contributors who provided the SARS-CoV-2 assemblies. We thank Dr. Adolfo Lara, Roger Sealy, Pamela Brown, Ryker Penn, and Yanlai Lai (Houston Health Department) for their assistance in sample collection and sequencing. The authors also would like to thank Lauren Bauhs, Russell Carlson-Stadler, Madeline Wolken, Kyle Palmer, Whitney Rich (Rice University) for their assistance in sample collection, processing, and analysis. This work was supported by the Houston Health Department. Funding sources for sequencing by the Houston Health Department were CDC ELC Enhanced Detection, CDC ELC Enhanced Detection Expansion, and CDC ELC Advanced Molecular Detection. N.S. is supported by the Ken Kennedy Institute Andrew Ladd Memorial Excellence in Computer Science Fellowship. N.S., Y.L., and T.J.T. were supported in part by the C3.ai DTI, Centers for Disease Control (CDC) contract 75D30121C11180, and P01-AI152999 NIH award. E.G.L. and L.B.S. were supported in part by the National Science Foundation (CBET 2029025), and seed funds from Rice University. K.B.E. was supported in part by the National Institute of Environmental Health Sciences, R01ES028819. T.J.T. was also supported by National Science Foundation grant EF-2126387.

## Author contributions
N.S., Y.L., E.G.L., and R.S. have contributed to data curation and formal analysis. N.S. and Y.L. have contributed to software development and validation. N.S., Y.L., T.J.T, and L.B.S. have contributed to the design and development of the methodology. L.H., K.B.E., L.B.S., and T.J.T. have contributed to the conceptualization of the study. All authors have contributed to preparing the original manuscript draft and reviewing and editing the manuscript.

## Competing interests
The authors declare no competing interests.
