## [Peer Review File · Nature Communications]

Reviewers' Comments:

Reviewer #1:

Remarks to the Author:

In their manuscript, the authors report a novel analysis technique they call QuaID for the identification of SARS-CoV-2 variants in RNA-Seq data from wastewater samples obtained from the Houston area. They compare their method to a competing tool, Freyja, and find QuaID to be more sensitive and capable of earlier detection of circulating variants.

The identification of viruses and viral variants is without doubt an important cornerstone of epidemiological surveillance, especially in the current scenario where less resources tend to be allocated to testing and sequencing of clinical samples. Tools developed in the context of the SARS-CoV-2 pandemic will also be of use in the future, when the focus may lie more tracking local and seasonal epidemics to provide a relatively quick and cheap early warning system. Thus, the method the authors developed is highly relevant.

That being said, some parts of the manuscript would benefit from clarification as well as some additional benchmark analyses.

Specifically:

- without actually running the pipeline, the results one gets from QuaID are a bit vague and should be described more in-depth in the manuscript. There's fields for "total allele frequency" (is this a pure sum of AFs, meaning that it is bounded by 0 and the number of quasi-unique mutations? In that case, is it correct that this value is not comparable across samples and variants?), "total QU count" (I assume that's the intersection of variant-specific QUs and those actually discovered in the sample?), "number of QU mutations possible" (this has me confused, is this the number specific for the strain? If yes, why is this zero for some strains in the example output?), and "fraction of QU sites covered" (again, I'm confused. Just because the QU isn't present doesn't mean it's site is not covered, why is this zero for some variants? And if the values are in the 70% range, why aren't more QUs detected? Omicron should have been the dominant strain in August of 2022, so why are there only 2 QUs, if >200 where possible and the coverage is >0.7?)
- I wasn't able to run the pipeline using the instructions on the Gitlab, pyvcf as a requirement seems to be missing (maybe consider adding a setup.py or a dump of your conda environment)
- I recognize that this was published shortly before the submission of the QuaID manuscript, but some comparisons with COJAC (<https://www.nature.com/articles/s41564-022-01185-x>) would be great, as this to my understanding has some similarities with QuaID (it uses V-Pipe, which in turn calls LoFreq, same as QuaID, and should thus be able to call indels). From my perspective, no in-depth benchmarking would be necessary, but a quick comparison and maybe an example run would certainly be interesting, especially since COJAC takes haplotypes into account. My guess would be that this might increase specificity at the cost of a loss of sensitivity.
- I like that the authors provide several simulations, and state their random seed. It is also not unsurprising to see that the results vary, especially for Freyja, which has fewer mutations to work with. That being said, I'm not fully convinced that purely random dropout is the best simulation strategy here. While I do not have experience with wastewater samples, I would assume that similar to any other RNA sample, their degradation is not random, but driven at least in part by exonucleases, leading to some parts of the viral genome (or, in the case of coronaviridae, subgenomic RNAs?) being more stable. At the same time, I wouldn't expect quasi-unique mutations to be spread randomly, as different portions of the viral genome should experience different selective pressure. Would it be possible to do an additional simulation with dropout probabilities being set to observed coverage probabilities?
- The manuscript mentions discriminating between "no detection" and "no coverage" scenarios and explicitly mention using the flanking region in the case of deletions. How is this handled for SNVs that are contained within a deletion? Would this be counted as "no detection"?
- I see two main limitations that both also affect competing tools, but should still be discussed: firstly, that only known strains can be detected. This might be remedied by reporting on mutations above a certain MAF that are not in the database, which is however not an addition I would request for the present manuscript. Secondly, that mutation thresholds are chosen based on frequencies in GISAID. The successful application of this surveillance method thus depends on enough (clinical?) sequencing samples being deposited. This isn't an issue at the moment, but may become one if awareness and funding decline in the future. Both issues should at least briefly be mentioned.

Minor comments:

- this is unrelated to the manuscript, but the authors may consider running the git garbage collection on the repository. The size of the object storage is quite excessive considering this is a relatively simple tool in terms of script length. The repository might also benefit from some tidier structure.
- there is little in terms of documentation for the tool. Specifically, the command line parameters should be described in more detail

Reviewer #2:

Remarks to the Author:

Sapoval et al present QuaID, an algorithm for detecting SARS-CoV-2 variants of concern in wastewater.

Detecting pathogens in wastewater is an important task which has been underscored during the pandemic. It is likely to be of ever more important in the next phases where population testing is not as widespread anymore and also for other pathogens.

Another method for this task is Freya, to which the authors compare, but other tools exists, e.g COJAC, <https://pubmed.ncbi.nlm.nih.gov/35851854/>

In the manuscript the authors claim that QuaID was capable of detecting Omicron up to three weeks earlier in Houston's wastewater than Freya. A main advantage of QuaID is the ability to include insertions and deletions in addition to base substitutions.

My main concern would be that the method requires further systematic evaluation to judge its future utility. While there appeared to be a greater sensitivity compared to Freja in detecting Omicron's emergence it's not clear whether this would be the case for any future variant. The author's conduct a basic SNV drop out simulation to support their specific claim, but I wonder how it would perform for future variants such as BA.4 and 5 and also some very recent lineages such as BA.2.75.2 or BQ.1.1, which are monitored by the SARS-CoV-2 surveillance community for their immune evasion capabilities.

Such a comparison should include varying the extent of divergence (perhaps expressed via substitutions and indels) and could be done by creating mixtures of very recent SARS-CoV-2 lineages. A comparison to further tools such as COJAC would also be beneficial.

LEGEND

- Responses to reviewers in red
- Changes to the manuscript in blue
- Location of inserted text in (parenthesis)
- Reviewers comments in black

Reviewer #1 (Remarks to the Author):

R1Q1. Without actually running the pipeline, the results one gets from QualD are a bit vague and should be described more in-depth in the manuscript. There's fields for "total allele frequency" (is this a pure sum of AFs, meaning that it is bounded by 0 and the number of quasi-unique mutations? In that case, is it correct that this value is not comparable across samples and variants?), "total QU count" (I assume that's the intersection of variant-specific QUs and those actually discovered in the sample?), "number of QU mutations possible" (this has me confused, is this the number specific for the strain? If yes, why is this zero for some strains in the example output?), and "fraction of QU sites covered" (again, I'm confused. Just because the QU isn't present doesn't mean it's site is not covered, why is this zero for some variants? And if the values are in the 70% range, why aren't more QUs detected? Omicron should have been the dominant strain in August of 2022, so why are there only 2 QUs, if >200 where possible and the coverage is >0.7?)

R1R1. We thank the reviewer for the helpful feedback. We have now updated the output description table on the GitLab page, and have added the new updated table to the Supplementary materials in the paper. We have also attached a copy of the updated table below for convenience.

Output column	Description
Date	Sample date
Plant	Sample (WWTP) name
Total AF	Sum of the allele frequencies of detected QU mutations (a floating point value in the range [0., Total QU count])
Total QU count	Count of detected QU mutations (i.e. the size of the set intersection between mutations designated as QU for the variant, and mutations occurring in a sample)
# QU possible	Total number of possible QU mutations for the given variant (if the selected time window in GISAID contains no sequences from the

	given variant or the inclusion/exclusion thresholds are set too stringently this value can be 0)
Fraction covered	Fraction of QU mutation sites with non-zero coverage (flanking coverage for indels). Note that this only checks the coverage of the site itself, and hence does not guarantee that a mutation was called (e.g. site with non-zero coverage, but less than 5x coverage or site with allele frequency below 0.02).
WHO name	WHO name for the considered VoC (e.g. Delta) or PANGO lineage name (e.g. BA.5)

Number of QU mutations possible is indeed 0 in the example output for all variants except Delta and Omicron. Since the simulated samples were marked as the August of 2022 and QualID was run with the default time window of 4 weeks up to sample collection date (meaning we only used GISAID sequence data that was collected at most 4 weeks prior to the sample collection date). For this time window GISAID doesn't contain any sequences belonging to the other lineages (with exception of Alpha and Gamma for each of which 1 sequence is present, and hence not considered for the analysis). Thus it is the expected behavior to see these variants with 0 possible QU mutations. If the user is interested in still tracking these variants they can modify the time window parameter to a larger value.

In case when the variant has no QU sites the coverage is considered 0, since no sites are undergoing the analysis. In case when a variant has QU sites, the interpretation is correct: this is the percentage of all sites that could contain a QU mutation and have non-zero coverage. We agree that this might create some confusion, since a variant caller might still be unable to call a mutation at a site even if the coverage is non-zero. For example, if the total depth of coverage at the site is low (typically below 10 reads) or if the allele frequency inferred is below 0.02 percent, additionally since we require simultaneous calls by iVar and LoFreq it's possible that only one of the variant calling softwares calls the mutation, and hence we do not consider it in the downstream analysis. Thus, it is possible to have a high percentage of sites with non-zero coverage, yet not a high number of QU detection events (this is usually the case with lower quality samples).

Finally, the samples presented in the example are not actual August 2022 Houston, TX samples. They are simulated reads based on a few sequences randomly sampled from GISAID database with additional noise added to closer mimic wastewater data. We chose to mark them as August 2022, to demonstrate the effect time window has on the possibility of detection of older variants (the 0 possible QU mutations case you have noted beforehand). We agree that the original presentation might have caused confusion, and hope that the expanded table is helpful in clearing this up. We have also added the table into the supplementary materials as Supplementary Table 2.

R1Q2. I wasn't able to run the pipeline using the instructions on the Gitlab, pyvcf as a requirement seems to be missing (maybe consider adding a setup.py or a dump of your conda environment)

R1R2. We have updated the GitLab by adding a conda environment dump (conda pack), as well as the set of updated instructions for running with demo data. We have also expanded the installation and running instructions to better capture realistic deployment scenarios. We asked two users uninitiated to running QualD and they were able to successfully install and run it from a fresh environment.

R1Q3. I recognize that this was published shortly before the submission of the QualD manuscript, but some comparisons with COJAC (<https://www.nature.com/articles/s41564-022-01185-x>) would be great, as this to my understanding has some similarities with QualD (it uses V-Pipe, which in turn calls LoFreq, same as QualD, and should thus be able to call indels). From my perspective, no in-depth benchmarking would be necessary, but a quick comparison and maybe an example run would certainly be interesting, especially since COJAC takes haplotypes into account. My guess would be that this might increase specificity at the cost of a loss of sensitivity.

R1R3. We appreciate the suggestion provided by the reviewer. We have conducted a brief comparison with the COJAC pipeline and identified the main commonalities and differences between the three methods (Freyja, QualD, and COJAC). We have summarized the results in the following table.

Tool	Considering indels	Key advantage
Freyja	No	Relative abundance inference
COJAC	Yes	High specificity
QualD	Yes	High sensitivity

Similarly to QualD and Freyja, COJAC requires a lineage definition for the execution. Unlike QualD and Freyja which construct such definitions automatically (by analyzing GISAID multiple sequences alignment directly via vdb in case of QualD, or by using mutation-annotated tree format from UShER generated trees in the case of Freyja) COJAC relies on the standardized variant definitions curated by Public Health England (PHE, https://github.com/phe-genomics/variant_definitions). As expected, high quality manually curated definitions, coupled with the same amplicon detection criteria of COJAC lead to very high specificity of detections. However, such an approach has to sacrifice some of its sensitivity in this case.

Specifically, we have evaluated COJAC's ability to detect the early Omicron events in Houston, TX wastewater data. QualD has recorded such events over the course of two weeks (three and

one weeks prior to the joint detection by Freyja and QualD, Figure 1A, C) with the first event consisting of detections of Omicron in 4 samples the week of 11/15/2021, and the second consisting of detections of Omicron in 10 samples the week of 12/02/2021. When run with the current definition (per PHE) of the Omicron variant of concern, COJAC confirms both detection events with all 4 samples on 11/15/2021 and 5 out of 10 on 12/02/2021 being marked for presence of Omicron variant. However, when COJAC is provided with the early definition of the Omicron variant per PHE (version date: 12-01-2021, git commit hash: 2f39758a37) which is relevant for the detection events described, only one sample on 12/02/2021 and none on 11/15/2021 are marked as containing Omicron variant by COJAC. Note, that QualD only considers GISAID data deposited prior to the sample collection date by default, and hence does not leverage more complete mutation profiles retroactively.

Thus, while retroactively COJAC confirmed our early detection events for presence of Omicron variant in Houston wastewater samples, when run with the data available at the time of detections it was unable to provide an early detection event. This result confirms our hypothesis that COJAC is a highly specific tool, which is great for scrutinizing and confirming detection events, but it does lack sensitivity towards recently emerged variants in the active pandemic setting.

To reflect the results of these additional analyses and our conclusions and recommendations for the potential users we have added the following paragraph to the discussion section of the paper:

We envision QualD to be one of several tools routinely employed in wastewater monitoring efforts. For example, QualD could be used in parallel with Freyja and COJAC to achieve high sensitivity for detecting emerging variants using QualD, relative abundance estimates for the dominant circulating variants using Freyja, and high specificity confirmations for detection events using COJAC.

R1Q4. I like that the authors provide several simulations, and state their random seed. It is also not unsurprising to see that the results vary, especially for Freyja, which has fewer mutations to work with. That being said, I'm not fully convinced that purely random dropout is the best simulation strategy here. While I do not have experience with wastewater samples, I would assume that similar to any other RNA sample, their degradation is not random, but driven at least in part by exonucleases, leading to some parts of the viral genome (or, in the case of coronaviridae, subgenomic RNAs?) being more stable. At the same time, I wouldn't expect quasi-unique mutations to be spread randomly, as different portions of the viral genome should experience different selective pressure. Would it be possible to do an additional simulation with dropout probabilities being set to observed coverage probabilities?

R1R4. We thank the reviewer for the suggestion. In order to address this concern we have conducted a more robust set of simulations in which real sample coverage was used as a guide for the dropout. Namely, we have taken the real wastewater samples collected the week of 12/02/2021 and extracted their coverage profiles by using samtools. Next, we pick and fix one

real sample coverage profile and for each sample in the simulated dataset we re-sample the mutations as follows: take p to be the allele frequency of a mutation in the simulated sample, then for the corresponding position take the real depth of coverage D and perform random Bernoulli trial with success probability p exactly D times. If as the result of this operation we obtain a mutation with allele frequency below 0.02 or the depth in the real sample is below 10 reads we remove the mutation call, otherwise we modify the depth and allele frequency with the ones obtained by resampling. This process would then be repeated for each of the selected templates yielding a total set of 24 (templates) x 104 (simulated samples) = 2,496 samples per random seed. In total we have evaluated more than 10,000 simulated samples.

We evaluated performance of Freyja and QualD on these extended simulations and found that there are 3 general scenarios occurring: (1) both tools perform well with QualD having earlier detection of the variants of concern analyzed (Alpha, Gamma, Delta, and Omicron); (2) both Freyja and QualD show degraded performance, for Freyja it manifests as a complete non-detection of Delta or Omicron variants across all samples in the batch, and for QualD it manifests as low sensitivity in detecting a variant of concern (i.e. detection only 4-6 weeks after the initial introduction); (3) in some cases QualD maintains high sensitivity of detection while Freyja exhibits a drop in sensitivity. Prototypical examples of each of the three scenarios described are shown below (Response figures 1-3). The full set of figures corresponding to this experiment have been added to the supplementary material (Supplementary Figures 8-31) out of those 11 follow the first scenario (both tools have robust detections, QualD has earlier detections by at least a week), 6 follow the second scenario (both tools show degraded performance), and 7 follow the third scenario (QualD mostly retaining its performance, while Freyja shows degraded performance).

Response figure 1. Y-axis shows relative abundance profiles inferred by Freyja above 0, and detection events reported by QualD (as dots) below 0. X-axis shows the week to which the simulated sample corresponds (sequencing data simulated from samples collected that week and deposited into GenBank under geographic location USA / Texas). In this scenario both QualD and Freyja show consistent detections of the variants of concern, with QualD detections occurring a week earlier than Freyja ones for Omicron (purple) and Gamma (pink) variants.

Response figure 2. Y-axis shows relative abundance profiles inferred by Freyja above 0, and detection events reported by QualD (as dots) below 0. X-axis shows the week to which the simulated sample corresponds (sequencing data simulated from samples collected that week and deposited into GenBank under geographic location USA / Texas). In this scenario both QualD and Freyja have degraded performance. Notably, QualD misses the detection of the first half of the Omicron wave, while Freyja reports no detections of Delta altogether.

Response figure 3. Y-axis shows relative abundance profiles inferred by Freyja above 0, and detection events reported by QualD (as dots) below 0. X-axis shows the week to which the simulated sample corresponds (sequencing data simulated from samples collected that week and deposited into GenBank under geographic location USA / Texas). In this scenario both QualD retains its performance relative to the ideal simulation, while Freyja has degraded performance for detection of the Delta variant.

R1Q5. The manuscript mentions discriminating between "no detection" and "no coverage" scenarios and explicitly mention using the flanking region in the case of deletions. How is this handled for SNVs that are contained within a deletion? Would this be counted as "no detection"?

R1R5. In case if a SNV is occurring inside a deletion region, and the deletion is present in the sample the result will be reported as "no coverage" for the SNV position, i.e. all mutations are considered independently, so from the SNV perspective this is a "no coverage" scenario, even if from the deletion perspective it can be a detection or non-detection event.

R1Q6. I see two main limitations that both also affect competing tools, but should still be discussed: firstly, that only known strains can be detected. This might be remedied by reporting on mutations above a certain MAF that are not in the database, which is however not an addition I would request for the present manuscript. Secondly, that mutation thresholds are chosen based on frequencies in GISAID. The successful application of this surveillance method

thus depends on enough (clinical?) sequencing samples being deposited. This isn't an issue at the moment, but may become one if awareness and funding decline in the future. Both issues should at least briefly be mentioned.

R1R6. We thank the reviewer for this suggestion. We have expanded the paragraph in discussion mentioning these limitations, as well as providing a brief outline for potential future work targeting putative novel lineages missing from the clinical data. Added text is also provided below for convenience.

Analogous to the other tools for variant detection, QualID depends on the database of known lineages, and hence cannot detect an emerging lineage that has not yet been designated as novel. However, the ability to track quasi-unique mutations for all known lineages combined with the variant calling in longitudinal samples can be extended in the future work to also enable detection of recurrent novel mutations and hence putative novel lineages. This is also important as the rate of clinical sequencing might decline in the future, leading to less densely sampled databases and hence potential for missing lineages in the available clinical sequencing data.

Minor comments:

R1Q7. This is unrelated to the manuscript, but the authors may consider running the git garbage collection on the repository. The size of the object storage is quite excessive considering this is a relatively simple tool in terms of script length. The repository might also benefit from some tidier structure.

R1R7. We appreciate the suggestion, and have subsequently cleaned up the repository and the file structure. We have also moved the larger data files, as well as the precomputed metadata and database files to OSF:

https://osf.io/tu6wh/?view_only=a975553294e34a90ab6ed9ac1fbc82bf to facilitate an easier option for the demo run.

R1Q8. There is little in terms of documentation for the tool. Specifically, the command line parameters should be described in more detail

R1R8. We have now addressed this by both providing short descriptions of parameters accessible via the --help command directly in the CLI, as well as a section in the README that describes each of the command line parameters. For key parameters that affect results the expected effect on the tool's performance is also discussed in the README.

Reviewer #2 (Remarks to the Author):

R2Q1. My main concern would be that the method requires further systematic evaluation to judge its future utility. While there appeared to be a greater sensitivity compared to Freja in detecting Omicron's emergence it's not clear whether this would be the case for any future variant. The author's conduct a basic SNV drop out simulation to support their specific claim, but I wonder how it would perform for future variants such as BA.4 and 5 and also some very recent

lineages such as BA.2.75.2 or BQ.1.1, which are monitored by the SARS-CoV-2 surveillance community for their immune evasion capabilities. Such a comparison should include varying the extent of divergence (perhaps expressed via substitutions and indels) and could be done by creating mixtures of very recent SARS-CoV-2 lineages. A comparison to further tools such as COJAC would also be beneficial.

R2R1. We thank the reviewer for the suggestions. We have conducted comparisons to COJAC with the results summarized below (excerpt from the response to Reviewer 1):

Similarly to QualD and Freyja, COJAC requires a lineage definition for the execution. Unlike QualD and Freyja which construct such definitions automatically (by analyzing GISAID multiple sequences alignment directly via vdb in case of QualD, or by using mutation-annotated tree format from UShER generated trees in the case of Freyja) COJAC relies on the standardized variant definitions curated by Public Health England (PHE, https://github.com/phe-genomics/variant_definitions). As expected, high quality manually curated definitions, coupled with the same amplicon detection criteria of COJAC lead to very high specificity of detections. However, such an approach has to sacrifice some of its sensitivity in this case.

Specifically, we have evaluated COJAC's ability to detect the early Omicron events in Houston, TX wastewater data. QualD has recorded such events over the course of two weeks (three and one weeks prior to the joint detection by Freyja and QualD, Figure 1A, C) with the first event consisting of detections of Omicron in 4 samples the week of 11/15/2021, and the second consisting of detections of Omicron in 10 samples the week of 12/02/2021. When ran with the current definition (per PHE) of the Omicron variant of concern COJAC confirms both detection events with all 4 samples on 11/15/2021 and 5 out of 10 on 12/02/2021 being marked for presence of Omicron variant. However, when COJAC is provided with the early definition of the Omicron variant per PHE (version date: 12-01-2021, git commit hash: 2f39758a37) which is relevant for the detection events described, only one sample on 12/02/2021 and none on 11/15/2021 are marked as containing Omicron variant by COJAC. Note, that QualD only considers GISAID data deposited prior to the sample collection date by default, and hence does not leverage more complete mutation profiles retroactively.

Thus, while retroactively COJAC confirmed our early detection events for presence of Omicron variant in Houston wastewater samples, when run with the data available at the time of detections it was unable to provide an early detection event. This result confirms our hypothesis that COJAC is a highly specific tool, which is great for scrutinizing and confirming detection events, but it does lack sensitivity towards recently emerged variants in the active pandemic setting.

We have also compared our results for detection of BA.4 and BA.5 lineages with the data reported by Freyja for a set of more recent samples from Houston SARS-CoV-2 wastewater surveillance project. Both tools are capable of detecting BA.4 and BA.5 lineages in the samples collected in summer 2022, as well as BA.2.75.* and BA.5.3.* recently emerged lineages. We

note that the simulations conducted by us (including the additional simulations with results reported in Supplementary Figures 8-31) have as their primary goal assessment of the effect of the sample degradation on the ability to detect known lineages. While the ability to detect significantly diverged and not yet reported lineages is out of the scope of this project and is a common limitation for all major tools considered.

Finally, we have expanded the discussion section by noting a limitation shared by all three tools (Freyja, COJAC, and QualD) for detecting previously unreported variants. Since QualD, Freyja and COJAC rely on the database of currently known lineages or lineage definitions, and hence depend on the corresponding genomes, neither of the tools will be able to detect a lineage not yet reported. However, for the lineages that have been reported in the database we should maintain high sensitivity, since the QualD quasi-unique mutation definitions are updated alongside the GISAID multiple sequence alignment and metadata updates. We have also elaborated further on the strengths and weaknesses of each of the tools, and emphasize that Freyja, QualD, and COJAC should be used in parallel. The added text from the discussion section is attached below:

Analogous to the other tools for variant detection, QualD depends on the database of known lineages, and hence cannot detect an emerging lineage that has not yet been designated as novel. However, the ability to track quasi-unique mutations for all known lineages combined with the variant calling in longitudinal samples can be extended in the future work to also enable detection of recurrent novel mutations and hence putative novel lineages. This is also important as the rate of clinical sequencing might decline in the future, leading to less densely sampled databases and hence potential for missing lineages in the available clinical sequencing data.

We envision QualD to be one of several tools routinely employed in wastewater monitoring efforts. For example, QualD could be used in parallel with Freyja and COJAC to achieve high sensitivity for detecting emerging variants using QualD, relative abundance estimates for the dominant circulating variants using Freyja, and high specificity confirmations for detection events using COJAC.

Reviewers' Comments:

Reviewer #1:

Remarks to the Author:

I would like to thank the authors for thoroughly addressing my concerns. The documentation of the tool, benchmarking, and comparisons with other tools have been improved, and the limitations are more clearly stated. Based on these changes, I see no further issues that need to be resolved before publication.

Reviewer #2:

Remarks to the Author:

The authors haven't been able to address my concerns unfortunately. In my previous comments I suggested a broader and more systematic comparison of informatics tools and also an investigation whether their approach is capable of detecting novel emerging variants. However, the only evidence the authors provide really pertains to the emergence of Omicron in Houston.

In the absence of such a comparison I'm afraid I cannot judge whether the authors' approach generally outperforms existing methods and also whether their approach will be capable of providing superior performance for future variants. These include for example the range of convergent immune escape lineages and also recombinants, including 'Deltacrons'. Based on past experience also further saltations may be possible and the tool of choice should facilitate the best possible detection of a broad range of new lineages.

In my eyes a systematic investigation would have entailed an appropriate quality metric (sensitivity, specificity) for each tool evaluated not only for historic data from different places, but also for emerging variants of the past year as well as putative novel variants - how likely is each tool to detect a new lineage that diverges by X substitutions and Y indels from the currently predominant variants?

LEGEND

Responses to reviewers in red
Changes to the manuscript in blue
Location of inserted text in (parenthesis)
Reviewers comments in black

Reviewer #1 (Remarks to the Author):

I would like to thank the authors for thoroughly addressing my concerns. The documentation of the tool, benchmarking, and comparisons with other tools have been improved, and the limitations are more clearly stated. Based on these changes, I see no further issues that need to be resolved before publication.

We thank the reviewer for the positive feedback. We are glad that additional benchmarks helped clarify the individual strengths and weaknesses of the tools. In particular, we decided to add a subsection to the Methods outlining the simulation protocols used to further improve benchmarking design.

Reviewer #2 (Remarks to the Author):

The authors haven't been able to address my concerns unfortunately. In my previous comments I suggested a broader and more systematic comparison of informatics tools and also an investigations whether their approach is capable of detecting novel emerging variants. However, the only evidence the authors provide really pertains to the emergence of Omicron in Houston.

R2Q1. In the absence of such a comparison I'm afraid I cannot judge whether the authors' approach generally outperforms existing methods and also whether their approach will be capable of providing superior performance for future variants. These include for example the range of convergent immune escape lineages and also recombinants, including 'Deltacrons'. Based on past experience also further saltations may be possible and the tool of choice should facilitate the best possible detection of a broad range of new lineages.

R2R1. We thank the reviewer for the suggestion of a more extensive comparison with the existing tools. We have expanded our benchmarking by generating 32,448 additional simulated read samples using coverage profiles of real wastewater samples as a guide. We have provided a quantitative comparison of the tools's performance on these new simulated data and added the following text and table to the manuscript in the Results section (p. 6).

Finally, we devised a simulation protocol (c) which resampled simulated sequencing reads based on a coverage profile guide from a real sample. In total we generated 32,448 simulated samples for this protocol and compared the performance of QualD, and Freyja.

Results of this comparison aggregated over all 32,448 samples are presented in Table 1. In this set of simulated experiments, QualD had the highest precision for all VoCs in consideration, further supporting our confidence in early detection of Delta and Omicron VoCs in Houston wastewater data. Freyja achieved a good balance between precision and recall as indicated by high F1 score. We hypothesized that the high precision of QualD was likely due to the combined variant caller output used in the pipeline. To test this, we conducted an experiment in which QualD used only iVar output for its analyses. As expected, we found that QualD's recall matched or in some cases surpassed that of Freyja at the cost of lower precision when using iVAR only (Supplementary Figure 32A, B).

Variant	Tool	Precision	Recall	F1 Score
Alpha	QualD	0.954	0.517	0.671
	Freyja	0.847	0.555	0.670
	COJAC	0.279	0.186	0.223
Delta	QualD	0.979	0.532	0.689
	Freyja	0.747	0.663	0.702
	COJAC	0.385	0.029	0.055
Gamma	QualD	0.999	0.343	0.511
	Freyja	0.686	0.414	0.516
	COJAC	N/D	N/D	N/D
Omicron	QualD	1.0	0.472	0.642
	Freyja	0.859	0.614	0.716
	COJAC	0.183	0.048	0.076

Table 1. Precision (number of true positives divided by the sum of true and false positives), recall (number of true positives divided by the sum of true positives and false negatives), and F1 score (harmonic mean of precision and recall) for QualD, Freyja, and COJAC on the data simulated under protocol (c). Bold font indicates highest F1 score, and N/D indicates no detections across all samples. Note that the results highlighted in red were not included in the main manuscript, since due to low recall values we considered COJAC not properly suited for VoC detection on the degraded environmental samples.

Response Figure 1. Recall of QualID, QualID (iVar), Freyja and COJAC on simulated data. Here QualID (iVar) denotes running QualID with variant calls provided by iVar instead of combining iVar and LoFreq calls.

Based on our current simulation based assessments COJAC is not well suited for early detection of emerging SARS-CoV-2 variants of concern in settings where sample quality can be low. However, we believe that COJAC is a very promising approach among the ones tested for detecting recombinant VoCs. We would like to point out that detection of recombinant lineages is not the goal of QualID, and we explicitly outline it as a current limitation, since we exclude recombinant genomes from our database construction step. We believe that detection of recombinants in pooled fragmented samples like wastewater remains a challenging problem, especially in the setting of degraded samples.

R2Q2. In my eyes a systematic investigation would have entailed an appropriate quality metric (sensitivity, specificity) for each tool evaluated not only for historic data from different places, but also for emerging variants of the past year as well as putative novel variants - how likely is each tool to detect a new lineage that diverges by X substitutions and Y indels from the currently predominant variants?

R2R2. To the best of our knowledge we have conducted the first large scale (more than 40,000 simulated samples in total across three simulation protocols: with 8,000 simulated samples analyzed under protocols (a) and (b) in the prior work, and more than 32,000 additional samples under protocol (c) analyzed in this response) benchmarking of VoC detection precision and recall for multiple tools used in wastewater SARS-CoV-2 surveillance. Furthermore, we have now provided a detailed description of our simulated benchmarking data in the Methods section of the paper under subsection Benchmarking tool performance with simulated data (p. 10). We also attach the added text below:

We have considered three simulation scenarios in order of increasing complexity: (a) random SNV dropout model, (b) coverage template based SNV resampling, and (c) coverage template based read resampling. In all three cases the base dataset was constructed from SARS-CoV-2 sequences deposited into NCBI GenBank before April 15, 2022, and collected between April 11, 2020, and April 15, 2022. Sequences were downloaded and grouped based on the collection week, yielding a total of 104 groups. For each group read data was simulated using ART short-read simulator. Each genome in the set representing a week was sampled with the same coverage, thus lineages with more representatives in each time frame yielded a higher amount of reads for that sample. Then the reads were processed identically to the regular pipeline processing. For the simulation scenario (a) after the variant calling was performed and results of LoFreq and iVar calls combined, we randomly retained 10%, 25% or 50% of all SNVs that were called. In the scenario (b) we augmented this process by introducing coverage templates. We selected 4 weeks of real data (corresponding to 06/14/2021, 07/12/2021, 11/15/2021, and 12/16/2021) and for each week we considered all 39 WWTPs and their corresponding sample coverage profiles. Given a coverage profile we performed a SNV resampling procedure directly on combined variant call files. The procedure consisted of changing the total depth of coverage at a SNV position to that in the coverage profile and if resulting total depth was below 10 the SNV was removed, otherwise the AF of the SNV was used as the probability of success in Bernoulli trial, and the fraction of successes from the total depth number of trials was set as the new AF for the SNV. However, since both scenarios (a) and (b) operated at the level of variant calls we could not use them for complete comparison. Thus, for the simulation scenario (c) we directly resampled reads in order to approximate coverage in the template profile. Template profiles were identical to the ones used in scenario (b). Given a coverage profile, we evaluated the coverage in positions 80 base pairs away from the 3' and 5' ends of the ARTIC v3 amplicons and used these values as target coverages. In order to build a sample, we sampled without replacement reads from the simulated data that overlapped at least one of target positions until either the target coverage, as defined above, was achieved or there were no more reads left to sample. The pipeline was then re-run with the modified read mapping files and the three tools benchmarked.

To the reviewers point, we agree that it can be of great value to the community to also integrate a forward looking simulation that incorporates evolutionary dynamics of SARS-CoV-2 in order to study the efficacy of different approaches for potential future

variants. However, developing and implementing such a benchmark is a complex undertaking that goes well beyond the scope of our current paper, often requiring a joint community collaboration and can entail years of effort (for example the Critical Assessment of Metagenome Interpretation effort

<https://www.nature.com/articles/nmeth.4458>,

<https://www.nature.com/articles/s41592-022-01431-4>, has resulted in development and improvements to the CAMISIM simulator

<https://microbiomejournal.biomedcentral.com/articles/10.1186/s40168-019-0633-6>).

In summary, the work outlined in our response to the original concerns serves as a major step in the direction of comprehensive benchmarks for wastewater surveillance tools, but we recognize that it has some limitations. We welcome follow up benchmarking studies that can be useful to further compare methods for detection of variants of concern directly from environmental samples.

Reviewers' Comments:

Reviewer #2:

Remarks to the Author:

The revision includes a comparison based on objective metrics as requested, which enables a better judgement of QuaID's performance.

Compared to Freya, QuaID had high specificity, albeit lower sensitivity (recall) and comparable F1 score for Alpha, Delta and Gamma. It has a higher sensitivity for detecting Omicron presumably because of its ability to detect indels.

The authors should reword their abstract accordingly, which currently states:

".. (ii) more sensitive VoC detection (tolerant of >50% mutation drop-out), and (iii) leverages all mutational signatures (including insertions & deletions)."

It seems that higher sensitivity is only given for Omicron, which contained characteristic indels. Sensitivity seemed lower for Alpha, Delta and Gamma.

Questions remain about the performance against current variants which mostly have substitutions. This should be discussed.

LEGEND

Responses to reviewers in red
Changes to the manuscript in blue
Location of inserted text in (parenthesis)
Reviewers comments in black

Reviewer #2 (Remarks to the Author):

The revision includes a comparison based on objective metrics as requested, which enables a better judgement of QualD's performance.

R2Q1. Compared to Freya, QualD had high specificity, albeit lower sensitivity (recall) and comparable F1 score for Alpha, Delta and Gamma. It has a higher sensitivity for detecting Omicron presumably because of its ability to detect indels.

The authors should reword their abstract accordingly, which currently states:

".. (ii) more sensitive VoC detection (tolerant of >50% mutation drop-out), and (iii) leverages all mutational signatures (including insertions & deletions)."

It seems that higher sensitivity is only given for Omicron, which contained characteristic indels. Sensitivity seemed lower for Alpha, Delta and Gamma.

R2R1. We have modified the text of the abstract to properly reflect the high precision that QualD achieves on the simulated benchmarks. We have removed the statement about overall higher sensitivity, as the simulated benchmarks do not support it. Modified abstract is provided below.

As clinical testing declines, wastewater monitoring can provide crucial surveillance on the emergence of SARS-CoV-2 variant of concerns (VoCs) in communities. In this paper we present QualD, a novel bioinformatics tool for VoC detection based on quasi-unique mutations. The benefits of QualD are three-fold: (i) provides up to 3-week earlier VoC detection, (ii) more accurate VoC detection (>95% precision on simulated benchmarks), and (iii) leverages all mutational signatures (including insertions & deletions).

R2Q2. Questions remain about the performance against current variants which mostly have substitutions. This should be discussed.

R2R2. We agree that understanding tradeoffs between the methods based on the underlying model of evolution (whether it is mostly SNVs or a mixture of indels and SNVs) is not fully investigated in the manuscript. To this extent we have added the following sentence to Discussion section (p. 6): However, assessing the impact that analysis of indels provides in the task of VoC detection requires more extensive evolutionary modeling.